# Deep learning for robust and flexible tracking in behavioral studies for *C. elegans*

**Kathleen Bates** [1,2☯], **Kim N. Le** [3☯], **Hang Lu** [1,2,3] *

**1** Interdisciplinary Program in Bioengineering, Georgia Institute of Technology, Atlanta, Georgia, United States of America, **2** School of Chemical & Biomolecular Engineering, Georgia Institute of Technology, Atlanta, Georgia, United States of America, **3** Wallace H. Coulter Department of Biomedical Engineering, Georgia Institute of Technology and Emory University, Atlanta, Georgia, United States of America

☯ These authors contributed equally to this work.
* hang.lu@gatech.edu

**Data Availability Statement:** All raw and annotated datasets and trained models are available as supplemental material via figshare (https://doi.org/10.6084/m9.figshare.13678705.v1). Supplemental figures and media can be found on figshare

## Abstract

Robust and accurate behavioral tracking is essential for ethological studies. Common methods for tracking and extracting behavior rely on user adjusted heuristics that can significantly vary across different individuals, environments, and experimental conditions. As a result, they are difficult to implement in large-scale behavioral studies with complex, heterogenous environmental conditions. Recently developed deep-learning methods for object recognition such as Faster R-CNN have advantages in their speed, accuracy, and robustness. Here, we show that Faster R-CNN can be employed for identification and detection of *Caenorhabditis elegans* in a variety of life stages in complex environments. We applied the algorithm to track animal speeds during development, fecundity rates and spatial distribution in reproductive adults, and behavioral decline in aging populations. By doing so, we demonstrate the flexibility, speed, and scalability of Faster R-CNN across a variety of experimental conditions, illustrating its generalized use for future large-scale behavioral studies.

## Author summary

Behavioral and ethological studies often rely on the ability to accurately track and process large amounts of behavioral recording in complex, heterogeneous environments and experimental conditions. Under these conditions, traditional image processing techniques may fail and prevent extraction of basic behavioral information, such as location and speed. We present an easy-to-use deep-learning-based tool to allow researchers to easily track and detect a commonly used model organism (*Caenorhabditis elegans*) in a variety of different environments. We examine behavior and movement throughout their lifespan, along with reproductive fecundity. We also provide a generalized, free, web-based tool for developing deep learning object detection models for any object of interest.

This is a *PLOS Computational Biology* Software paper.

(https://doi.org/10.6084/m9.figshare.13681675.
v4). The pipeline is available via github (https://
github.com/lu-lab/frcnn-all-in-one). Further details
of the imaging system used for the developmental
assays can be found at our GitHub: https://github.
com/lu-lab/mi-pi.

**Funding:** This study was funded by US NSF
(1764406) and US NIH (R01AG056436,
R01GM088333) grants to HL, US NIH F31
fellowship to KB (F31GM123662) and US NSF GRF
to KL (DGE-1650044). The funders had no role in
study design, data collection and analysis, decision
to publish, or preparation of the manuscript.

**Competing interests:** The authors have declared
that no competing interests exist.

## Introduction

Ethology has been crucial in the fields of neuroscience, genetics, and aging [1–4]. This rings true even in the simplified *C. elegans* model, which has been used to probe a variety of ethological questions [5–10]. In these experiments, it is extremely valuable to robustly and accurately track and measure the behavior of *C. elegans* on a large scale. The ability to collect large-scale behavioral data has significantly improved throughout the years. While many behavioral assays consist of manually recording small populations of animals on agar plates under a stereomicroscope, recent automated methods have drastically increased the variety of biological questions researchers can explore. Ranging from multi-camera systems, to time-shared imaging systems, to low-cost imaging systems, advances in hardware have allowed users to more easily obtain large amounts of raw behavior video [11–15]. In addition to improvements in data acquisition, there have been advances in culture methods, which enable the exploration of more complex environmental conditions. These range from individual "arenas", which allow the tracking of populations with individual level resolution, to microfluidic devices, which allow for precise spatiotemporal environmental control [12,16,17]. While these technological advancements have enabled the ability to explore complex behavior relevant to neuroscience and aging, this increase in behavioral recordings and data shifts the bottleneck to the analysis of large-scale image datasets. This is especially crucial for images taken in heterogeneous environments, such as those in more complex, naturalistic conditions.

One of the major challenges in analyzing behavioral data is the detection and identification of the object of interest, especially under a variety of imaging and environmental conditions. While there are many existing image processing tools that are currently used to detect, identify, and subsequently analyze the behavior of worms [18–20], there are unmet needs. These tools often use traditional image processing methods, such as background subtraction, thresholding based on the color or intensity of the object, or the use of morphological features (such as size), to detect and identify the object of interest. For example, in the popular worm tracker Tierpsy Tracker [18], users manually optimize parameters based on experimental conditions and are subsequently able to extract behavioral data from their dataset. WormPose [20] leverages traditional image processing techniques for segmentation of animals, coupled with machine vision to resolve complex shapes and postures without the need for human annotation or labeling. With these segmentation and tracking tools, users can extract a variety of informative behavioral phenotypes, such as size, speed of movement, and the posture of individuals. However, with the advent of more complex experimental setups that introduce more heterogeneous experimental or environmental conditions, it is not straightforward to adapt these traditional methods to robustly and accurately detect objects of interest. For example, in conditions with low or uneven imaging contrast, basic thresholding based on intensity values may not be accurate. If animals or the objects of interest move only subtly, background subtraction cannot be used to easily differentiate between the object of interest and the background of the image. Even with advanced tools like WormPose, these challenges make it difficult to obtain an accurate initial training set in order to resolve complex postures. Additionally, if there is a wide range in morphological properties, such as the dramatic size change of animals during development, it is difficult to rely on traditional morphological features such as size as a method of identifying objects of interest. Thus, coupled with the increased scale of behavioral datasets, there is a need for a robust, flexible, and facile method to detect and identify worms that would be able to work across a variety of different experimental conditions, with minimal user input.

To address this problem, we turn to deep learning, which has emerged as a powerful data-driven tool for object detection. Deep learning has previously been used to extract pose estimates in animals, which has enabled experimenters to measure complex phenotypes, ranging

from gait patterning in flies to examining odor trail tracking in mice [21–23]. However, many of these existing tools use complex architectures and are often difficult to set up, and require large, difficult-to-annotate training sets, and greater computational power to classify each individual pixel. In contrast, object detection, which identifies bounding boxes of objects, can be used as a measure to extract meaningful behavioral phenotypes, requiring less computational power and annotation time than deep learning methods for pixel classification, while providing greater accuracy than heuristic segmentation methods. While there are many deep learning object detection methods, the Faster R-CNN architecture is a widely-used method that uses region proposal networks (RPN) coupled with convolutional neural networks (CNN) to extract the location (in the form of bounding boxes) and estimated likelihood for each detected object [24]. It is one of the top performing object detection methods, as measured by the mean average precision (mAP) of detections on the standardized COCO dataset [25]. Compared to other CNN methods with equivalent or higher mAP, the Faster R-CNN architecture is less computationally costly and thus advantageous for large volumes of data. Further, the Faster R-CNN architecture has been tested in a wide range of applications, ranging from vehicle and pedestrian detection to malarial detection via cell classification [24,26]. In the context of animal detection, it has been used in proof-of-concept applications to detect cattle in animal husbandry conditions, as well as detecting a variety of species in camera traps in the wild [27–29].

Here, we implement Faster R-CNN to identify and locate worms across a variety of different conditions without extensive user input. We find that after the initial training, the deep learning model quickly and accurately detects objects of interest. We demonstrate its flexibility across a variety of different recording platforms and imaging modalities. We also demonstrate its ability to detect worms across a variety of different ages (from L2 to death), showing its flexibility across different body sizes and movement levels, and illustrate how it can be used to extract useful behavioral metrics and trends to give insights into biological questions, such as egg laying, development, and behavioral decline in aging. Finally, we provide a web-based pipeline (https://github.com/lu-lab/frcnn-all-in-one) for testing our trained models with novel data and to enable other researchers to annotate and train their own object detection models with novel data and classes.

## Results

To illustrate the difficulty in identifying and tracking objects in complex conditions, we examined three common experimental set-ups. The first tracks an individual worm from the L2 larval stage to Day 1 of adulthood on an agar plate seeded with food (**Fig 1A**). During this period, the worm was free to roam throughout the field of view, and animals were imaged through early adulthood. This type of low-magnification imaging setup is common for long-term and high-throughput imaging [19,30–33], as well as lifespan imaging [11]. For small L2 animals, a major challenge in tracking is the small size of the individual (starting at around 360 μm in length) and differentiating it from the background despite the low contrast and low magnification of the image. Tuning heuristics-based image processing tools to optimize for the small size and low contrast of young animals leads to further challenges as the worm grows (**S1A–S1E Fig**). The contrast from background improves as the worm develops; however, other subtle changes in background (such as eggs or tracks formed on lawns) may be identified as animals when using heuristics tuned for young animals (**S1C–S1E Fig**). These heuristics are also highly dependent on environmental and imaging conditions. In the case that changes in illumination or environment are an integral part of the experiment, this leads to an inability to process data without further tuning (**S1E Fig**). Together, these challenges make processing developmental behavior data a labor-intensive task.

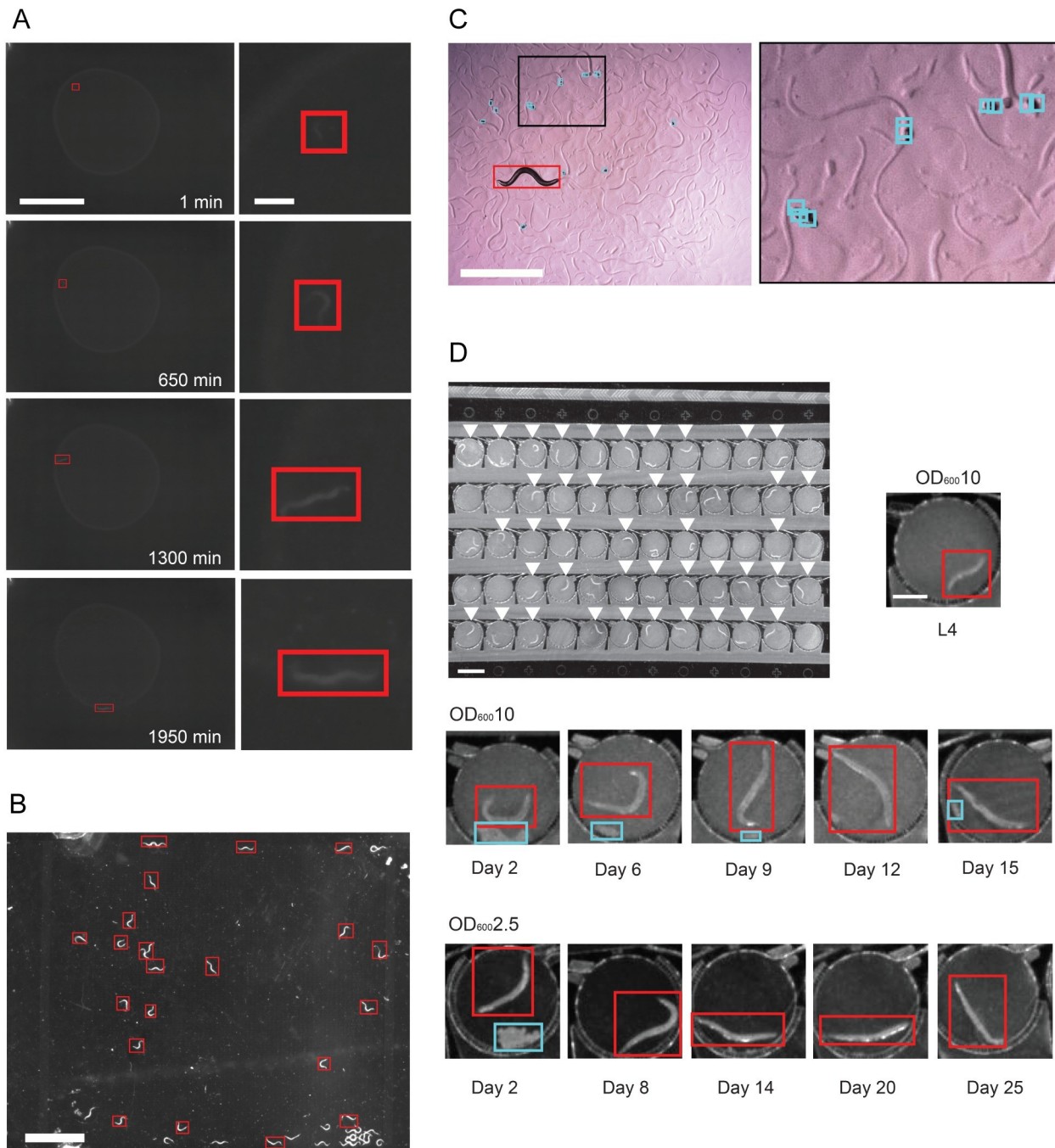

**Fig 1. Deep learning can be used to detect objects in a variety of complex environments.** A. Tracking an individual worm throughout its developmental period (from L2 to Day 1 adulthood). *(left)* Images of the plate over time. Scale bar is 5mm. The red box is the Faster R-CNN detection of the tracked worm. *(right)* Zoomed in image of the worm detected by the WoP Faster R-CNN model. Scale bar is 0.5mm. B. The WoP model applied in a different experimental set-up (adult worms in a microfluidic pillar array chamber). Worms detected by the WoP Faster R-CNN model are marked with red bounding boxes. Scale bar is 3mm. C. An egg-laying adult on an agar plate, with detected eggs boxed in blue and detected worm boxed in red. All detections made with egg-finder Faster R-CNN model. Scale bar is 1mm. *(right)* Zoomed in image of a cluster of eggs detected by the model. D. Tracking worms through their adult lifespan. *(top)* Microfluidic chamber array that cultures individual worms within each chamber (white arrows indicate chambers with single worms). White scale bar is 1.5mm. *(bottom)* Zoomed in images of individual worms under different food levels across the entirety of their adult lifespan. Worms detected by the WiCh Faster R-CNN model are boxed in red. Egg clusters detected by the WiCh Faster R-CNN model are boxed in blue. Scale bar is 0.5 mm.

The second experimental system measures trends in worm fecundity over time. An adult worm was allowed to roam across a seeded agar plate and freely lay eggs (**Fig 1C**). Due to the small size of eggs (~50 μm) [34], low magnification and contrast, and the tendency for eggs to be laid in clusters, it can be both time and labor intensive to manually count the number of eggs over time and mark their spatial location. Further, some studies may involve egg-laying behavior in different environments (e.g. on or off bacterial lawns) where imaging conditions and contrast pose significant problems in identifying these objects and distinguishing them from other objects in the field of view. In contrast to animals, the immobility of eggs also prevents the use of background subtraction as a useful tool. These practical constraints make it difficult to track fecundity and other egg-laying phenotypes at a large scale.

The third example is tracking the behavior and movement of individuals during the aging process. Worms were cultured within a microfluidic chamber array (**Fig 1D, top)** from the L4 stage to their death. Individuals were longitudinally monitored and their behavior was recorded intermittently throughout their lifespan under a variety of different food concentrations (**Fig 1D, bottom rows)**. While the size of the worm and the contrast are better than those for young animals during development, there are two inherent challenges. First, as before, the environment is heterogeneous—within the chamber there are often moving objects aside from the worm (such as debris or eggs laid by the individual), making it difficult to accurately identify and detect the location of the worm even in instances with high levels of movement. Second, there are low levels of movement as the worm ages and eventually dies, making it difficult to identify the worm through traditional image processing techniques that rely on movement. These challenges are cumbersome to address when using traditional image processing tools that require tuned user parameters. For instance, when the parameters are chosen for an individual video (**S2A and S2B Fig, top row**), it fails to accurately identify and detect the location of worms of the same age and under the same environmental and imaging conditions (**S2A and S2B Fig, bottom rows**). When the worm is in its reproductive period, the presence of laid eggs that cluster together in the chamber can cause misidentification and inaccurate segmentation of the worm (**S2A Fig**). When the worm is aged and only performs small, subtle movements the algorithm often truncates or misidentifies the location of the worm entirely (**S2B Fig**).

For both the on-plate and on-chip conditions, traditional detection and tracking methods are unable to robustly identify the location of the worm due to the worms' complex surroundings. While classic segmentation methods based on heuristics can provide posture information that CNN object detection methods cannot, the need to specifically tune parameters for a wide range of videos makes it challenging to deploy these methods at large scales in each of the demonstrated experiments. Thus, there is a need for a quick and generalizable method of identifying and tracking objects of interest in challenging imaging conditions such as these.

To address these challenges, we implemented Faster R-CNN, a deep learning network with high precision in object detection, including small objects. From an existing Faster R-CNN model pre-trained on the COCO image dataset, we tuned the model using our respective behavioral data sets [35,36]. For each of the three different experimental conditions, we trained a separate Faster R-CNN model. For the condition with developing worms cultured on an agar plate we created the worms-on-plate (WoP) model. For the WoP model, we annotated worms with bounding boxes in 1,122 randomly chosen and representative images from a much larger dataset and trained the model with 1,008 of these, holding out the remainder as a test set to evaluate the model. To measure worm fecundity, we created the egg-finder model. We annotated eggs and worms in 127 images and used 114 of these to train the model, with the remainder used to for evaluation of model performance. For both WoP and egg-finder models we used exclusively images of N2 animals. Lastly, for the more specialized condition of worms

cultured within the microfluidic chambers, we created the worms-in-chamber (WiCh) model. For the WiCh model, we annotated eggs and worms in 5,176 images, with 4,658 of these used to train and the remainder to evaluate the model.

When we qualitatively examined the bounding box output of these trained models, we found that many of the failure cases using traditional methods were resolved (**Figs 1 and S1 and S2**). In the WoP dataset, worms were identified accurately in both very low-contrast images when worms were very small as well as in much higher-contrast images later in life (**Fig 1A**). We also found that this model could detect worms in very different imaging conditions than those with which it was not trained, as for the animals imaged behaving in liquid media in **Fig 1B** (also see **Supplemental Movie 3 in** https://doi.org/10.6084/m9.figshare. 13681675.v6). Despite the contrast being extremely different in **Fig 1B** compared to **Fig 1A**, most of the single animals in the field of view are successfully detected. Those animals that are undetected are likely a result of a training set that does not include any images from a similar imaging set-up as well as the high confidence score (80%) specified to ensure a low false positive rate when locating worms. This is highly significant as it indicates the model is much more generalizable compared to heuristic techniques and thus more widely usable in real applications. In the egg-finder dataset, we found that eggs were identified well despite their small size and tendency to cluster together (**Fig 1C**). Notably, when we applied the egg-finder model to publicly available videos from the Open Worm Movement Database, we were able to detect both worms and eggs (**Supplemental Movie 1 in** https://doi.org/10.6084/m9.figshare. 13681675.v6). Finally, in the more specific WiCh dataset, worms were identified accurately, even in the most food-dense, low-contrast settings and at later ages where worm movement is reduced (**Fig 1D**). In addition, clusters of eggs could be accurately identified, making it possible to differentiate active worm movement from passive movement of the egg cluster (**Fig 1D**).

Next, we quantitatively evaluated our models to ensure their accuracy (**Table 1**). A common metric used in object detection is average precision (AP), which uses the overlap between actual bounding boxes and those predicted by the model at varying confidence thresholds to evaluate the model performance. An AP equal to unity would indicate perfect predictions. For the WoP model, we were able to obtain an average precision of 0.969. AP for the youngest animals (66 test images) was 1.0 with our test set, compared to the oldest animals (22 test images), for which false positives reduced the AP slightly to 0.876 (**S3 Fig**), making the model robust across age without further tuning. For our egg-finder model, the worm AP was 0.932 and the egg AP was 0.74 (**S3 Fig**). While the average precision for eggs is not as high as for the other objects we detected, we found that it was able to identify 79% of the eggs in our test set and that the sensitivity of the model for our test set was 0.84 (**S3 Fig**). Conservative identification of eggs by the model likely stems from the size of eggs making the overlap threshold (intersection over union) of detections and ground truths particularly sensitive, as well as the intensity of the eggs being similar to other image features such as the tracks created by animal movement, and the occlusion of eggs by each other. In practice, we found this model worked well to

**Table 1. Detection results across different confidence thresholds on the development, egg laying, and aging detection models using Faster R-CNN.**

| Model | Category | Average Precision (AP) @ threshold 0.5 | Average Precision (AP) @ threshold 0.1 | Average Precision (AP) @ confidence threshold 0.01, IoU threshold 0.3 |
|---|---|---|---|---|
| Development | Worms (all ages) | 0.969 | 0.969 | 0.969 |
| Egg counting | Eggs | 0.398 | 0.430 | 0.740 |
| Aging | Worms (all ages) | 0.998 | 1.00 | 1.00 |

identify trends in egg-laying phenotypes in further experiments (see below, **Fig 2**), and was able to detect eggs in data collected by other labs (**Supplemental Movie 1 in** https://doi.org/10.6084/m9.figshare.13681675.v6). It also may be possible to improve the average precision by using more training data. For the WiCh model, we obtained an AP of 0.998. This model is also robust across different conditions within the dataset, ranging from different ages, contrast levels, and objects of interest (**S3 Fig**), with the models detecting not just worms but also the laid eggs within the field of view (AP of 0.932).

In addition to the models being flexible across conditions without the need for additional parameter tuning, the inference time for each image is short (~131 ms/frame on our equipment). This is significant because in practice, these generalizable strategies can reduce the time and effort it takes to quantify new data. For instance, in cases with large sets of data under varied conditions, traditional hand annotation or the implementation of user-tuned parameters would require excessive amounts of time and manual labor. In contrast, deep learning enables users to analyze large behavioral datasets in a more efficient manner.

We next put the algorithm to a real use-case—monitoring egg-laying phenotypes of *C. elegans*. Egg-laying rate is indicative of health [37], evolutionary fitness [38–40], and is also important in understanding the regulatory mechanisms of the reproductive circuit [37,41–43]. However, the small size (approximately 50 μm) and large number of eggs (about 300 per adult hermaphrodite) makes measuring fecundity a challenging task. Manually counting eggs is time-intensive, but it is often still the method of choice because existing automated methods for egg detection are very sensitive to imaging conditions, requiring high image uniformity and often high magnification [43–45]. Likely due to the challenging nature of the task, several popular software packages for worm behavior quantification do not include methods to track egg-laying [18,19,46]. Other egg-counting methods rely on specialized cytometry to count eggs as they are flushed from liquid culture [47,48]. Another factor that makes counting eggs difficult is the burst-like timing of egg-laying events [49–51]. Many egg-laying events in close temporal proximity causes eggs to cluster, making identification of individual eggs challenging, and occlusion of the egg by the worm's body also makes identifying and locating eggs challenging. Further, the bacterial lawn that adults feed on becomes highly textured as adults crawl on it, which makes identifying eggs and animals significantly more challenging for image thresholding methods. To determine whether the egg-finder model would resolve these issues, we applied it to count and locate eggs (**Fig 2**). We collected images of individual day 1 adult worms and the eggs they laid at two time points, 2 and 5 hours after transferring animals onto plates within the typical range of bacteria density for individual culture. While we qualitatively found that our model did not successfully identify every egg, the agreement between manual egg counts and those from our model was significant (**Fig 2A and 2B**). Even when illumination was uneven across the field of view, when eggs were laid on the lawn, and when eggs clustered together, the model produced accurate results (as exemplified in **Fig 2A**). On occasion, recently laid eggs were detected after worms moved away from them due to occlusion of the egg by the worm. While the model may produce a short lag in detection of eggs because of this, it may still be valuable for evaluating trends in time between egg-laying events. This robustness in the face of highly variable conditions indicates that this method is a much faster alternative to manual counting of eggs that can capture important trends. This method is especially well suited for large datasets and movies where manual annotation of every frame would be prohibitively difficult or when imaging conditions like textured bacterial lawns prevent thresholding techniques from performing well.

Next, we showed that we can apply the algorithm to accurately estimate two egg-laying rates for each animal in early adulthood from images of individual animals' brood at two time points (**Fig 2C**). These results match previously reported egg-laying rates of about 4–10 eggs/

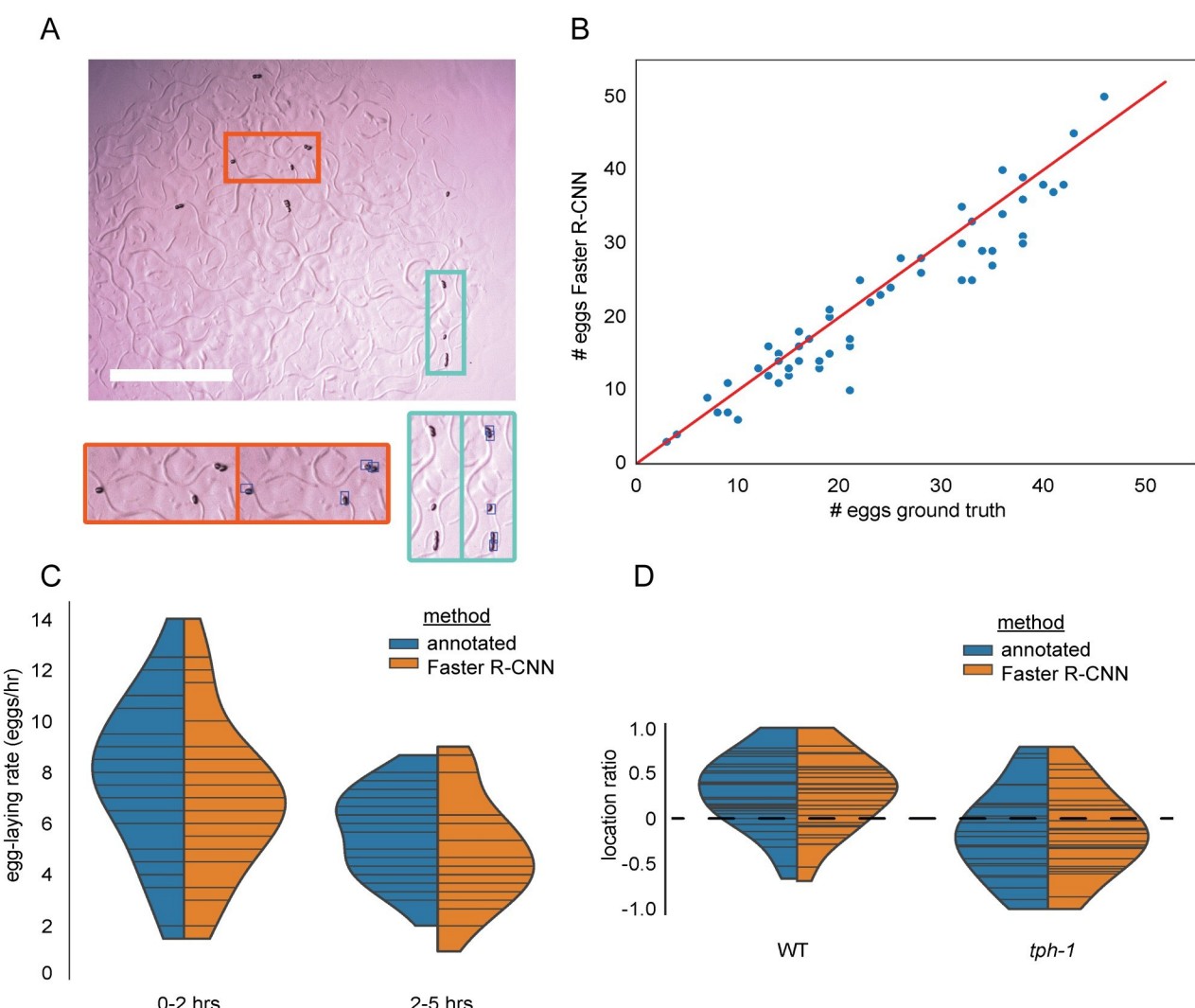

**Fig 2. Automated egg detection using Faster R-CNN.** A. Representative image from egg-finder dataset. Blue boxes overlaid on the right inset images indicate egg detections using the egg-finder Faster R-CNN model at the confidence score threshold of 0.01. Scale bar is 1mm. B. Agreement between manual egg count and egg-finder Faster R-CNN model egg count. Eggs laid by individual animals were counted manually and using the Faster R-CNN model with a confidence score threshold of 0.01. The agreement between these two counts was measured at two time points for n = 29 individual animals. Distributions were compared using the Kolmogorov-Smirnov 2-sample test and found to be not significantly different (KS statistic is 0.155, p value is 0.491). C. Egg-laying rate distribution for 29 animals at 2 time points. Horizontal bars within each distribution represent egg counts for individual animals. Distributions were compared using the Kolmogorov-Smirnov 2-sample test. The egg-laying rate counted manually at timepoint one was significantly different from the egg-laying rate counted manually at timepoint two (KS statistic is 0.448, p value is 0.0053). The egg-laying rate counted using the egg-finder Faster R-CNN model with a confidence threshold of 0.01 at timepoint one was significantly different from the egg-laying rate counted using the egg-finder Faster R-CNN model with the same threshold at timepoint two (KS statistic is 0.448, p value is 0.0053). For each timepoint, the Faster R-CNN egg-laying rate distribution was compared to the manually counted distribution, and for both timepoints, the KS test statistic was determined to be 0.2069 with a p value of 0.5141. D. Egg-laying preference for N2 (n = 16) and *tph-1* (n = 13) animals. Horizontal bars within each distribution represent egg counts for individual animals. Positive scores indicate a higher propensity to lay eggs on the lawn compared to off the lawn. Negative scores indicate a higher propensity to lay eggs off the lawn (see Materials and Methods for calculation formula). Distributions were compared using the Kolmogorov-Smirnov 2-sample test. N2 vs. *tph-1* distributions were different at a significant level for both manual counts (KS stat is 0.473, p-value of 0.0018) and egg-finder Faster R-CNN model counts (KS stat is 0.459, p-value is 0.0028).

hr [50]. The distribution of egg-laying rates for the manual and Faster R-CNN model egg counts was statistically indistinguishable at both timepoints, whereas comparing the Faster R-CNN model egg count distributions and manual egg count distributions between time-points showed a significant difference. We noted that egg-laying rates decreased over time,

which we suspect is due to transferring animals from a crowded growth plate to individual plates. This change in the experienced levels of oxygen, carbon dioxide, and food may promote a higher egg-laying rate until the individuals habituate to the new environment and begin to deplete food [37,49,52]. The ability to detect this difference using the egg-finder model demonstrates that Faster R-CNN can be used to identify biologically relevant phenotypes in a less time-intensive way than manually counting eggs.

In addition to identifying a rate phenotype, we used the egg-finder model to identify a spatial phenotype using the food-sensing mutant *tph-1*. These animals are known to spend a greater fraction of time in a roaming state and are also slower to pause upon encountering food [13,53,54]. Based on this and anecdotal evidence (Dhaval Patel, personal communication), we expected that *tph-1* animals would lay more eggs off of bacterial lawns compared to wild type animals. To examine whether this was the case, we defined an egg location preference score such that a greater number of eggs laid on the bacterial lawn would result in a positive preference score, while a negative preference score would indicate that a greater number of eggs were laid off the bacterial lawn. We found that the distribution of preference scores for WT and *tph-1* animals was consistent with *tph-1* animals having a lower preference for laying eggs on the bacterial lawn, and that the two distributions were significantly different both when counted manually and when counted using the Faster R-CNN model (**Fig 2D**). This example demonstrates that the model can distinguish eggs and capture important trends on or off the lawn regardless of lighting, contrast, and despite the width of adult animal's tracks being almost the same size as an egg. Further, we applied the egg-finder Faster R-CNN model to movies from the publicly available Open Worm Movement Database and found that we were able to successfully detect both eggs and worms in these movies without additional training (**Supplemental Movie 1 in** https://doi.org/10.6084/m9.figshare.13681675.v6). We also note that since the Faster R-CNN algorithm does not take movement into account, animals with locomotion defects like those of *tph-1* or more severe locomotion mutants should be detected similarly well. Overall, the egg-finder model performed well across both datasets in different imaging conditions and was able to uncover the same egg-related phenotypes as a human annotator in our own dataset again suggesting that Faster R-CNN models can replace manual labor particularly for large-scale datasets and movies with complex imaging conditions.

In addition to quantifying aspects of behavior through endpoint snapshots, we reasoned that Faster R-CNN could also be used to track animals over time. Estimating animal linear and angular velocity is a useful indicator of the animals' behavior state (e.g. dwelling/ roaming) as well as potentially an indicator of health when observed over sufficiently long timescales [16,55]. Using the dynamic location of worm bounding boxes detected using our WoP model, we evaluated whether it was possible to obtain accurate movement measurements. We compared the centroids of detected bounding boxes obtained from our model to the centroids of hand-annotated postures at 5 series of time points over about 2 days of worm development (**Fig 3A and 3B**). Throughout the ~ 2 days of observation, the object detection model was able to accurately identify worms, with the smallest animals an average of 124 pixels (~0.12 mm$^2$) in area (**Fig 3A**, left column). This timescale includes development from late L2 stage through to adulthood. We found that the bounding box centroids detected were typically close (0.212 mm +/- 0.197 mean+/- standard deviation) to the centroids of the hand-annotated bounds of the worm, indicating that this method can provide accurate worm locations at discrete time points (**Figs 3A**, right column, and **S6B–S6D**). To test whether we could also measure motion accurately with this method, we used time points a minute apart and calculated motion between the bounding box centroids and hand-annotated worm shape centroids at these time points. The difference between these two motion measurements was an average of 0.126mm

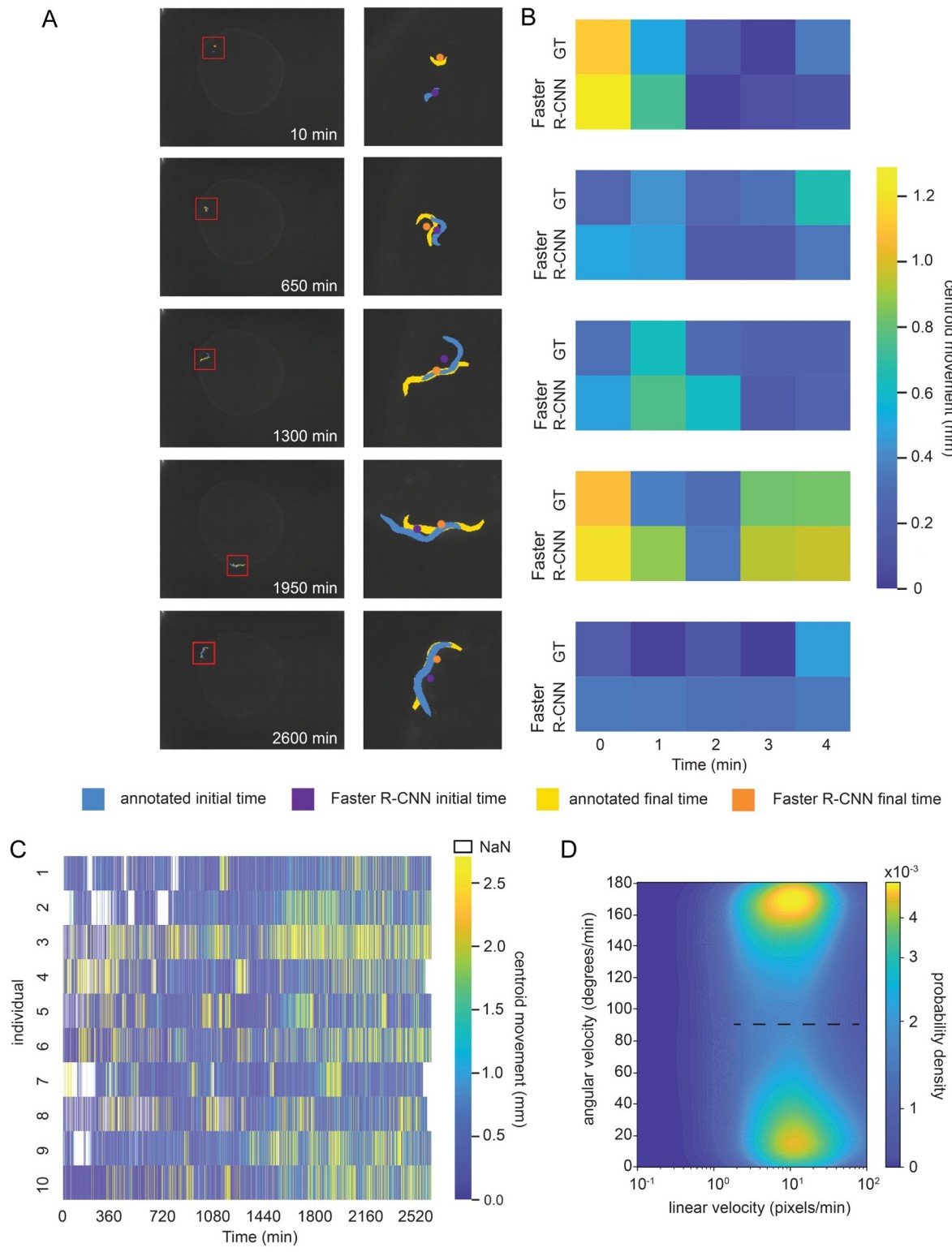

**Fig 3. Tracking behavior in development.** A. A single N2 animal tracked throughout development at 5 example time points. At left, the full-size image is overlaid with a red box highlighting the worm location for each timepoint. At right, the boxed portion of the image is overlaid with the manual annotation at the example timepoint (blue) and five minutes later (yellow), as well as the centerpoint of the WoP Faster R-CNN model's detected bounding box at the example timepoint (purple) and five minutes later (orange). B. Heatmap comparing distance travelled by worms calculated from manual annotations and WoP Faster R-CNN model detections. For each timepoint in A and

the subsequent five minutes, the distance travelled by the worm is calculated from manually segmented animals and from the WoP Faster R-CNN detections. The manual distance is calculated from the Euclidean distance travelled at the centroid of the segmented animal for each pair of time points, while the Faster R-CNN distance travelled is calculated from the Euclidean distance between the centroid of WoP Faster R-CNN bounding boxes for each pair of time points. C. Heatmap of centroid movement measured using the Faster R-CNN model for 10 individuals over the course of development from late L2 stage through adulthood. D. Histogram of Faster R-CNN derived movement speeds for the example animal in A and B.

+/- 0.083mm (mean +/- std) and motion trends were clearly replicated between the hand-annotated shape centroids and the bounding box centroids (**Figs 3B** and **S6C**). We further compared peak and mean velocity in our hand-annotated dataset to peak and mean velocity computed using the WoP Faster R-CNN model centroids. We found that both peak and mean velocities were very similar, with differences on the order of pixels in 5MP images (**S1 Table**). Similarly, we found that we were able to very accurately recapitulate the linear and angular velocity and peak and mean velocities of a publicly available dataset using our WoP Faster R-CNN model [13] (**S4 and S5 Figs and Supplemental Movie 3 in** https://doi.org/10.6084/m9.figshare.13681675.v6). Even without knowledge of the precise pixels that comprise the worm, the extents of the worm can be used as a rough measurement of movement and speed.

We next examined how this approach could be used to track behavior continuously over development. Behavior during development is individualistic and can affect long-term behavior and neuropeptide signaling in *C. elegans* [13]. The dual challenges of the small size and low contrast of young animals have previously imposed stringent hardware requirements to ensure high-quality images (e.g. at higher magnification), therefore limiting the scalability of long-term developmental experiments. In contrast, the WoP model can extract worm position despite low image quality and the extreme variation in the size and contrast of developing worms. We collected time-lapse images at 1-minute intervals of 10 animals over a 2-day period from late L2 stage to adulthood, and detected worm bounding boxes for each image (**Fig 3C and Supplemental Movie 2 in** https://doi.org/10.6084/m9.figshare.13681675.v6). The magnitude of motion increased over time, and likewise, we found that the size of the detected bounding box grew approximately 5-fold over time as the animals developed (**Figs 3C and S6A**). We next examined whether our centroid data could be used to identify roaming and dwelling states, which are an indicator of satiety and which are influenced by neuromodulators. *C. elegans* spends the greater portion of its time moving at slower speeds while dwelling and a small portion of its time searching for other food sources (roaming) [50,53,56]. We calculated the linear and angular velocity for each animal and found that while there was little separation in linear velocity between the two states, there was a striking split in angular velocity that was consistent with the roaming and dwelling state separation in other datasets at the same 1-minute sampling rate (**Figs 3D and S7 and Supplemental Movies 4 and 5 in** https://doi.org/10.6084/m9.figshare.13681675.v6, and see **Materials and Methods**). While the lower sampling rate introduces some loss in the classification, we emphasize that using the Faster R-CNN WoP model recapitulates both linear and angular velocity well when applied to independently collected data with higher sampling rates (**S5 Fig**). Thus, this technique can enable researchers to infer high-level information about the animal's behavioral state from the limited information provided by bounding box identification.

Next we tested how well the model can be used to track motion in a realistic biological discovery context: examining behavioral decline in the aging process. Behavior and movement are common methods to gauge the health and physiological age of an individual [12,14,16,55,57]. For these experiments, it can be challenging to accurately measure how the movement of individuals changes throughout their entire lifespan due to complex environmental conditions and the large scale of the data. As a specific example, we examined wild-

type individuals cultured in a microfluidic device, allowing us to identify and track individuals throughout the entirety of their adult lifespan (**Supplemental Movie 6 in** https://doi.org/10.6084/m9.figshare.13681675.v6). Due to the size of each chamber (~1.5 mm in diameter), as the worm grows, the extent of movement becomes limited, making common metrics (such as tracking the distance traveled by the centroid of the segmented worm) unable to clearly provide insights into the decline of movement over time. As a result, to gauge movement we examined the normalized sum of the difference in pixels across frames for segmented individuals (**Fig 4A**). As the worm ages and its movement declines, the difference across frames decreases as well for the overall population (**Fig 4B**). However, although this method provides useful insight into the behavioral decline of the individual, it can be difficult to accurately obtain the properly segmented worm in large datasets. This is due to issues with background contrast, the presence of eggs in the chamber, and the low mobility of older worms. Furthermore, the process typically requires large amounts of manual parameter tuning (to account for the different sizes of worms as they age, changing levels of movement with age, and variation across the individuals) and substantial computational time to segment and extract features of interest. The need for manual tuning and intensive computational resources makes it difficult to scale this method for large sets of behavioral information.

Faster R-CNN can serve as a quick and accurate alternative to gauge behavioral decline with aging across a population. By tracking the bounding box locations of the worm detected by the WiCh model and measuring the IoU (intersection over union) of the detection bounding boxes across the video, we can get a rough metric of movement. Young, highly active worms have little to no bounding box intersection across frames, while older, slower moving worms have increasing levels of intersection across frames (**Fig 4C**). To examine how movement changed with time we examined a movement score (1 –IoU), and observed individual decline in movement, as well as a population-level behavioral declines with age (**Fig 4D**). Not only were we able to view similar patterns of movement decline to that observed with segmented frames, we were able to do so on a larger scale with minimal processing time (~131 ms/frame).

Further, we wanted to examine whether this movement score could discern how perturbations influence decline in movement with age. Dietary restriction (DR) is an evolutionary conserved perturbation that has been shown to modulate aging [58–60]. We examined the movement score of worms cultured under constant DR ($OD_{600}$2.5) starting at Day 2 of adulthood and were able to demonstrate that the worms under lower food levels had a statistically significant difference in behavioral decline compared to worms cultured at higher food levels ($OD_{600}$10), trends observed in a prior study (**Fig 4E**) [14]. In addition, to verify the performance of the model on the dataset, we also validated that the detected bounding boxes from the WiCh model were comparable to the bounding box of the hand annotated, segmented worms (**S8 Fig**). This exercise demonstrates that motion quantitatively estimated by Faster R-CNN can be used as a quick metric to track and examine behavioral decline within an aging population.

## Discussion

Processing big sets of behavior data remains a major challenge currently facing large-scale ethological studies. As a model organism, *C. elegans* is well-poised as a subject for large-scale investigation, but typical computer vision analysis pipelines may still be insufficient in complex imaging conditions, where the animal itself may change size, contrast with background, or where inhomogeneities in the environment lead to failure of heuristic models. Here we have shown that applying Faster R-CNN object detection models to identify, count, and track

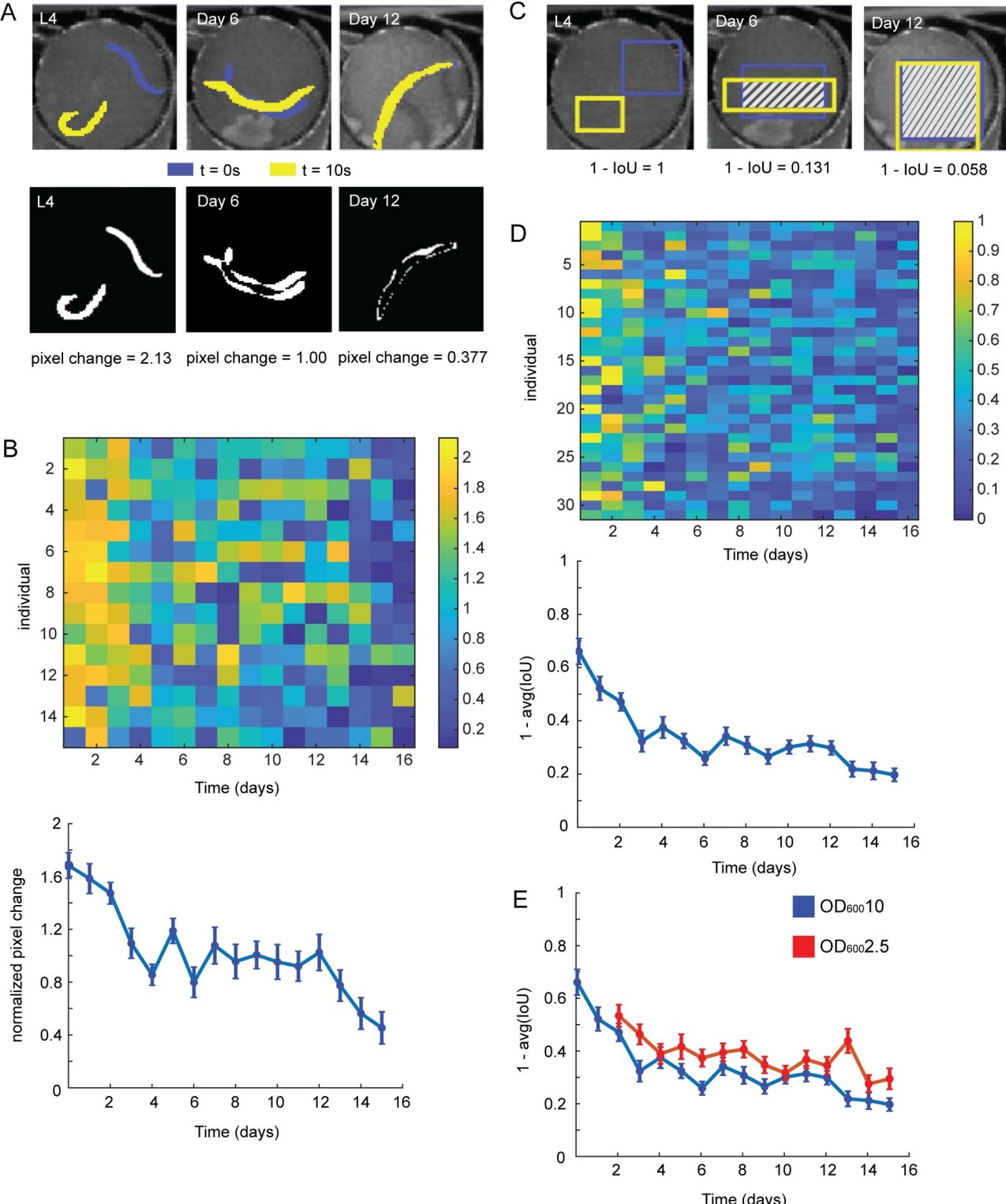

**Fig 4. Tracking behavioral decline in aging.** A. (top) Representative image of movement as the worm ages. The posture at the beginning of the video is shown in blue. The posture at the end of the video is shown in yellow. (bottom) Absolute difference image of the frames shown on the top. Pixel change values are the sum of the absolute difference image normalized by the average worm size in the video. B. (top) Heatmap of individual behavior decline (via pixel changes) over time (n = 15). (bottom) Average pixel change values over time from L4 to Day 15 of adulthood. Error is plotted as SEM. C. Representative image of the bounding boxes obtained using Faster R-CNN as the worm ages. The box found at the beginning of the video is shown in

blue. The box found at the end of the video is shown in yellow. The IoU of the two boxes is shaded. D. (top) Heatmap of individual movement (1 –IoU) from L4 to Day 15 of adulthood (n = 31). Individuals are cultured in $OD_{600}10$. (bottom) Average movement decline over time. Error is plotted as SEM. E. Average movement decline over time for individuals cultured in high levels of food ($OD_{600}10$ in blue) and individuals cultured in low levels of food ($OD_{600}2.5$ in red). Error is plotted as SEM. Movement for $OD_{600}10$ and $OD_{600}2.5$ is significantly different via Kolmogorov-Smirnov 2-sample test (p = 0.0003).

behaviors in challenging environments is a fast and flexible alternative to more traditional analysis methods. We first demonstrated this method's applicability in identifying eggs and estimating egg-laying rates and spatial distribution of eggs. We were also able to show that this method was effective in tracking movement of animals ranging from L2 stage through the end of life, providing high-level behavior state information as well as information relevant to animal health. This effectively includes many applications in behavioral and aging studies using *C. elegans*.

Compared to other conventional methods, we found this deep learning method to be significantly more generalizable across a variety of experimental conditions. It requires no specialized hardware or dedicated imaging set-up and, once trained, we found the Faster R-CNN models worked very well, even under conditions for which they had not been trained (**Supplemental Movies 1 and 3 in** https://doi.org/10.6084/m9.figshare.13681675.v6). In comparison, other methods (including other machine-learning image classification tools like Ilastik [61]) needed additional manual tuning for even slightly different lighting conditions, differently sized animals, or changing contrast levels (**S9 Fig**). Here we have not used animals smaller than the typical young L3 worm; this is purely due to the magnification limitations of our imaging systems. As we show with the egg-finder model, at higher magnifications smaller objects such as eggs can certainly be detected. While for small datasets, the time investment for annotating images and training the model may be high, for large datasets with imaging variability the high detection accuracy and elimination of video-by-video or frame-by-frame hand-tuning represents a significant gain. For datasets that require extensive human correction, the accuracy of manual annotation degrades unpredictably over time as attention wanes; in comparison, machine learning models are predictably biased based on the content of the training set. These advantages, combined with the high speed of processing with a GPU (on average 7.6 fps), makes the deep learning approach a very good alternative to more conventional methods.

While we have shown that this deep learning method is very generalizable, there are inevitably cases where the existing trained models (WoP, egg-finder, or WiCh) are unable to robustly detect objects of interest when imaging conditions are sufficiently different, or for example if animal morphology is significantly changed by mutations. In order to enable other labs to adapt this methodology for unique datasets, we have also developed an easy-to-use web-based tool that enables users to annotate, retrain, and evaluate their dataset (accessible from https://github.com/lu-lab/frcnn-all-in-one). The pipeline is easy to use, with naïve users unexperienced in Python or coding languages being able to independently use the tool. There is a low barrier-to-entry for use, with no dependence on paid, commercial software (such as MATLAB) and no requirement for downloading external programs. We also include further documentation on how to quantitatively validate trained models for users that do not have experience with deep learning methods. While we found that the Faster R-CNN architecture suited our accuracy and speed requirements, this pipeline can also be used to train models using other object detections architectures that achieve faster inferencing speeds with somewhat lower accuracy. Using the Faster R-CNN architecture with our pipeline, we found that training on as few as 10–20 annotated images for several hours provided very high-quality tracking results (**Supplemental Movie 7 in** https://doi.org/10.6084/m9.figshare.13681675.v6).

This time investment will likely pay off for large-scale datasets and datasets where imaging conditions create challenges for classic thresholding methods.

We also showed that even without segmenting images and extracting the posture of the individual and more complex behavioral phenotypes as many traditional methods do, we were still able to extract information about animal and egg location, size, linear and angular velocity, and animal behavior state that could be used to identify biologically meaningful phenotypes. These simple methods have a lower computational cost than that of segmentation, and in cases where precise knowledge of posture is not necessary, this method provides a fast and environmentally robust estimation of relevant metrics. Additionally, there are related deep learning approaches that provide semantic segmentation, such as a Mask R-CNN model, which would enable extraction of posture information. The application of this model is a natural next step, although the annotation necessary to train such a model is significantly more intensive. To train a Faster R-CNN, only 2 points are required to define a box containing the object of interest, making annotation fast. This compares favorably to segmentation methods like Mask R-CNN that require annotating many points to approximate the outline of objects.

In addition to the ability to accurately identify and track animals quickly, the success of this deep learning method in extreme imaging conditions suggests that this method can be used to push the current limitations in quantifying animal behavior in ethologically relevant environments. Researchers face a trade-off between performing assays in naturalistic environments and the ability to extract more information from more uniform and controlled environments. Deep learning methods such as this one may provide us with a greater ability to extract the necessary information from richer environments with greater ethological relevance.

## Materials and methods

### *C. elegans* maintenance

*C. elegans* strains were maintained under standard conditions at 20˚C unless otherwise noted [62]. Strains used in this work include N2 and QL101[*tph-1(n4622) II*].

### Plate assays

To prevent animals from leaving the microscope field of view (FoV), we prepared special plates. Palmitic acid has been demonstrated as an effective barrier for worms in behavior experiments [63]. It is typically applied as a solution in ethanol to a standard plate and the ethanol is allowed to evaporate off. However, it is hard to deposit in a controlled way due to the palmitic acid solution wetting the agar. We used an ethanol-sterilized piece of PDMS as a negative to prevent a 10 mg/mL palmitic acid in ethanol solution from wetting the center of a 5cm NGM plate, allowing the ethanol to evaporate for at least 30 minutes before removing the PDMS with tweezers. These plates were subsequently seeded with 10 μl (developmental experiments) or 5 μl (egg-laying experiments) of OP50. Plates used in developmental experiments were incubated at room temperature for about 24 hours to allow a thin lawn to form and were stored at 4˚C until an hour before use. For egg-laying experiments, plates were seeded approximately 2 hrs before transferring animals onto plates.

For developmental assays, adult animals were bleached to obtain eggs. Eggs were allowed to hatch and larvae allowed to reach L1 arrest by agitating eggs overnight in M9 buffer. L1s were then pipetted onto an unseeded NGM plate and single animals were pipetted onto the prepared seeded palmitic acid plates. These plates were then parafilmed and incubated at 20˚C until animals reached late L2 stage (20 hours after plating), when each plate was placed on a Raspberry Pi-based imaging system. The Raspberry Pi imaging system used a Raspberry Pi v3 Model B (Raspberry Pi Foundation) with official Raspberry Pi touchscreen (Raspberry Pi

Foundation) and a Raspberry Pi Camera Module v2 (Raspberry Pi Foundation) with no additional lens to capture images at minute time intervals. Darkfield illumination was provided by an LED Matrix (Adafruit), with a center circle of LEDs dark and the surrounding matrix illuminating animals with red light. Further details of this imaging system can be found at our GitHub: https://github.com/lu-lab/mi-pi. Developmental experiments lasted 44 hours, at which point worms have typically reached sexual maturity and plates were removed from imaging systems.

For egg-laying assays, gravid day 1 adults animals were picked onto prepared palmitic acid plates and plates were imaged at 2 and 5 hours at 1.6x on a stereomicroscope (Leica M165 FC) using a 1.3 MP CMOS camera (Thorlabs DCC1645C) with a 0.5x coupler.

### *C. elegans* on-chip culture

Synchronized L4-stage wildtype animals were loaded into a worm chamber array microfluidic device. Microfluidic devices were fabricated from polydimethylsiloxane using standard soft lithography techniques and sterilized by autoclaving.

Worms were cultured at 20˚C in *E. coli* (HB101) spiked with Pluronic F-127 (0.005%), carbenicillin (50μg/ml), and kanamycin (50μg/ml) to prevent the risk of bacterial aggregation and contamination during long-term culture. The bacteria was at a concentration of $OD_{600}$ 10 to prevent any harmful effects of dietary restriction on the developmental process. The bacteria also contained 5uM of C22, which interrupts eggshell formation and results in non-viable progeny. At Day 2 of adulthood, worms were then shifted to 25˚C and to the desired food level. Individuals shown in **Fig 4** were maintained at $OD_{600}$ 10 unless otherwise stated. Individuals in **Fig 1** were cultured at $OD_{600}$10 and $OD_{600}$2.5. We used an average flow rate of approximately 15μL/min across all conditions. See prior work for more details. [14]

### Training Faster R-CNN network

For the egg detection and aging model, we used TensorFlow GPU (v 1.14) to train the model. For the developmental tracking model, we used TensorFlow CPU (v 1.14). For all models except the model trained using the web-based pipeline we used the pre-trained 'Faster_rcnn_inception_v2' model from the Tensorflow 1 model zoo (https://github.com/tensorflow/models/blob/master/research/object_detection/g3doc/tf1_detection_zoo.md) and fine-tuned it with our data sets of interest. We trained the models and processed the images on a system with an Intel(R) Xeon(R) CPU E5-1620 v4 processor and a NVIDIA Quadro M4000 GPU.

For the egg detection model, images were taken of a mixed population of wild-type worms on a seeded plate at 1.6x magnification on a stereomicroscope (Leica M165 FC) using a 1.3 MP CMOS camera (Thorlabs DCC1645C) with a 0.5x coupler. We manually annotated 127 images of worms and eggs using the labelImg Python package. Images were randomly split into training and testing sets using a rough 90/10 split (114 images for training, 13 test images).

For the developmental tracking model, images were taken as described for plate developmental assays above. 1,122 images were randomly selected from a large set of developmental imaging data (> 10,000 images taken on 8 different imaging setups) and annotated using the labelImg Python package. This annotated image set was divided into 1,008 images for training and 114 images for testing, roughly a 90/10 split. No images used in training or testing overlap with image data evaluated in **Fig 4**.

For the aging model, videos were taken at an acquisition rate of 14 fps using a 1.3 Megapixel monochrome CMOS camera (Thorlabs DCC1545M camera) coupled with a 10X close focus zoom lens (Edmund #54–363). Each video was 10 seconds in length. Illumination was provided by a set of concentric red LED rings (Super Bright LEDs 60 and 80mm LED Halo

Headlight Accent Lights) to reduce the amount of blue light exposed to the worm. Videos were sampled evenly throughout the lifespan of individuals in food levels of $OD_{600}10$ and $OD_{600}2.5$. We manually annotated 5,176 frames of worms and, if present, eggs using the labelImg Python package. Images were randomly split into training and testing sets using a rough 90/10 split (4658 images for training, 518 test images).

For the model trained with our web-based pipeline, we annotated 14 frames total and used 12 frames to train a Tensorflow 2 Faster R-CNN model pre-trained with the COCO image dataset for 2 hours. We used the resulting model to perform the detections in Supplemental Movie 7 in https://doi.org/10.6084/m9.figshare.13681675.v6.

**Faster R-CNN model characterization.** For each of our annotated datasets, we evaluated how well our model performed by calculating precision and recall as well as average precision. Precision is a measure of the false positive rate, as calculated by $Precision = \frac{TP}{TP+FP}$, where TP is the number of true positives and FP is the number of false positives. Recall is a measure of the false negative rate, as calculated by $Recall = \frac{TP}{TP+FN}$, where FN is the number of false negatives. Average precision is the integral of the precision recall curve for a set of images that have ground truth bounding box annotations as well as model predictions. To determine whether detections by the models were true positives, false positives, or false negatives, we used a measure of the overlap of detections and ground truth known as intersection over union (IoU), calculated as $IoU = \frac{|GT \cap P|}{|GT \cup P|}$, where GT is the bounding box of the ground truth and P is the bounding box of the prediction. An IoU $\geq 0.5$ is counted as a true positive and an IoU $< 0.5$ as a false positive for our worm detections, with the IoU threshold lowered to 0.3 for egg detections in our egg-finder model. A false negative is counted when a ground truth annotation has no overlap with a detection by the model. Once all images with ground truth annotations are evaluated in this way, the maximum precision at each recall level is used to interpolate between points of the precision-recall plot.

The AP, recall, and precision for the egg-finder Faster R-CNN model was evaluated using a score threshold of 0.01. The AP, recall and precision for the WoP Faster R-CNN model and the aging model was evaluated using a score threshold of 0.5. Mask annotation and centroid computation for ground-truth movement comparisons of data in **Fig 3** was collected using MATLAB.

**Evaluation of egg-laying phenotypes.** Images of individually cultured animals were collected at 2 hrs after transfer onto individual plates and again at 5 hrs after transfer. For each image, a human curator manually counted eggs and identified them as being on or off the bacterial lawn. The Faster R-CNN model was also used to detect eggs in each image at a confidence threshold above 0.01, and eggs were manually identified as being on or off the bacterial lawn. We used these detections to overlay bounding boxes on each image and manually classified each detection as on or off the bacterial lawn. As the arena where the worm was able to roam was larger than the microscope field of view, images were tiled to ensure all eggs laid by each individual were captured. In cases where images overlapped with one another, double-counted eggs were subtracted from the overall count. The egg-laying preference score was calculated as follows: $preference\ ratio = \frac{eggs_{on} - eggs_{off}}{eggs_{on} + eggs_{off}}$.

**Evaluation of roaming and dwelling during development.** To establish that roaming and dwelling information can be extracted, we used [13] as a benchmark to establish the classification criteria of different states. This is because our data were collected at lower temporal frequency (images are collected every minute in our developmental dataset) while the data from [13] were collected at higher frequency but can be down-sampled to match the sampling frequency of our data. Here, we used their entire collection of centroid information for 123 N2 animals through development, which was originally collected at 3fps. We followed the method

used in Stern *et al.* (2017) [13] for determining whether animals were roaming or dwelling at any given timepoint to use as ground truth (**S7A Fig**). We then artificially reduced the temporal frequency of their data to 1 minute to match the temporal frequency of the data in our developmental dataset (**S7B Fig**). We further tested sampling frequencies between 1s and 60 minutes to visualize how the linear and angular velocity were affected by a variety of sampling rates (**Suppelemental Video 5 in** https://doi.org/10.6084/m9.figshare.13681675.v6). This video shows the gradual loss of separation between the roaming and dwelling states when sampling frequency decreases. At a 1-minute sampling rate, however, the separation persists.

Because of the clear split in angular velocity when using a 60s sampling frequency and little difference in linear velocity at this frequency, we determined how well we could predict the ground truth by simply splitting the reduced sampling rate data into angular velocities greater or less than 90 degrees/min. With this simple method we achieved 79% accuracy on a dataset of >70 million frames. Comparatively, when we applied linear discriminant analysis (LDA) to the 60s frequency data, we were only able to increase accuracy 5%, indicating that the split at 90 degrees/s is likely to be close to as good as any other linear split for centroid data collected at 1 minute intervals. We thus applied the same 90 degree/min angular velocity split to the developmental data to classify roaming and dwelling states (**Supplemental Video 4 in** https://doi.org/10.6084/m9.figshare.13681675.v6).

**Evaluation of behavioral decline in aging.** Images of worms were segmented through hand annotations using Ilastik. To calculate the pixel difference as the worms age we used $\sum \frac{|img1-img2|}{(img1+img2)/2}$ where img1 was the initial segmented frame of the video and img2 was the final segmented frame of the video. The IoU of the bounding boxes were calculated using the built-in MATLAB function bboxOverlapRatio. The 1-IoU metric was found by looking at the overlap between the bounding box found in the first frame and the bounding box found in the last frame of the video.

## Supporting information

**S1 Fig. Limitations of traditional image processing techniques in developmental monitoring.** A. Detection of animals throughout development using the trained WoP Faster R-CNN model or a common tool that uses traditional segmentation (Tierpsy Tracker). Tierpsy Tracker parameters were manually tuned to detect the animal in the example image in A, and not re-tuned for analyzing the same animal at later timepoints in B-E. Successful segmentation of the worm by Tierpsy Tracker is denoted by a white arrow, with non-worm segmentations marked by red arrows (*middle column*). Worm detections using the WoP Faster R-CNN model are bounded by a red box (*left column*). All worm detections shown reached a confidence threshold of 0.99. The animal in A is detected by both the WoP model and Tierpsy Tracker, but other non-worm objects are identified based on the optimized Tierpsy Tracker segmentation. B. The same animal as in A at a later timepoint. The animal is identified using the WoP model, but not identified by Tierpsy Tracker. C. As contrast improves, the same animal is detected by both Tierpsy Tracker and the WoP model, but the segmentation parameters as optimized for small, low-contrast animals also pick up non-worm objects. D. Once the animal becomes a gravid adult, the animal is identified by both Tierpsy Tracker and the WoP model, but eggs and tracks in the bacterial lawn increase the number of non-worm segmentations by Tierpsy Tracker. E. Illumination changes increases the number of non-worm segmentations by Tierpsy Tracker, while the WoP model is still able to identify the animal and no other non-worm objects.
(PDF)

**S2 Fig. Limitations of traditional image processing techniques in aging populations.** A. Detection of a young worm using traditional techniques (Tierpsy Tracker) or the trained WiCh Faster R-CNN model *(top row)* Successful detection of a worm via Tierpsy Tracker. Parameters for traditional techniques were manually tuned for this specific video. *(bottom rows)* Detection of worms (under the same age and environmental condition) using the same parameters as before. Detection errors are highlighted by red arrows. Red boxes show detection location via WiCh Faster R-CNN model. B. Detection of an old, slow moving worm using traditional techniques (Tierpsy Tracker) or the trained WiCh Faster R-CNN model *(top row)* Successful detection of a worm. Parameters for traditional techniques were manually tuned for this specific video. *(bottom rows)* Detection of worms (under the same age and environmental condition) using the same parameters as before. Errors and misidentification are highlighted by red arrows. Red boxes show detection location via WiCh Faster R-CNN model.
(PDF)

**S3 Fig. Precision-recall curves for the detection models.** A. Precision-recall curves for worm detection in the WoP model with confidence threshold of 0.5. Precision recall curve for all worms (left), L2-L3 stage animals (middle), and adult animals (right). B. Precision-recall curve for the worm detection (left) and egg detection (right) in the egg-finder model with confidence threshold of 0.01. The intersection over union used to determine true positive detections for eggs was 0.3, compared to 0.5 for worms. C. Precision-recall curve for the overall worm detection (left) and egg detection (right) in the WiCh model with confidence threshold of 0.5 (top row). Precision-recall curves for the worm at varying stages in the lifespan with confidence threshold of 0.5 (middle row), and across different food levels/contrasts with confidence threshold of 0.5 (bottom row).
(PDF)

**S4 Fig. Accurate centroid tracking in other datasets.** A. Comparison of X centroid coordinates from Stern *et al.* (2017) [13] to Faster R-CNN WoP model detections of the same data. Gap in Faster R-CNN detections at approximately 9 minutes occurs when animal collides with edge of circular arena. B. Comparison of Y centroid coordinates from Stern *et al.* (2017) [13] to Faster R-CNN WoP model detections of the same data. Gap in Faster R-CNN detections at approximately 9 minutes occurs when animal collides with edge of circular arena.
(PDF)

**S5 Fig. Accurate linear and angular velocity analysis with other datasets.** A. Comparison of binned angular velocity over time from animal and timepoints of animal in **S4 Fig** (data from Stern *et al.* (2017) [13]) and Faster R-CNN WoP detections of the same data. Data was binned by first smoothing angular velocities using a 10s moving average window (as in Stern *et al.* (2017) [13]), then thresholding the data into low and high angular velocities. The angular velocity values represent the average angular velocity of the low and high angular velocity data for each data set independently. B. Linear velocity of both datasets vs. time for animal and timepoints shown in **S4 Fig**. Linear velocity was calculated in the same way from both datasets, then smoothed with a moving average window of 10s, and finally by removing outliers. C. Scatterplot comparing Faster R-CNN WoP centroid velocities for animal and timepoints in **S4 Fig** to Stern *et al.* (2017) [13] ground truth velocities. Some structure is apparent because consecutive timepoints are likely to have correlated velocity, and in the case of the Faster R-CNN detection, correlated errors.
(PDF)

**S6 Fig. Accuracy of tracking in development.** A. Change of bounding box size over time using WoP Faster R-CNN for animal depicted in 4A. Smoothed using a moving window

average over 10 time points (10 minutes). Where no animal is detected, line is not connected. B. Density histogram of distances between box centroid as detected by WoP Faster R-CNN model and centroid of annotated worm shape for animal in 4A (n = 30 time points). C. Density histogram of difference between movement calculated from annotations and movement calculated from WoP Faster R-CNN model for animal in 4A (n = 25 time points). D. Centroid movement as calculated by the WoP Faster R-CNN model vs. centroid movement as calculated from manually segmented animals for animal in 4A (n = 25).
(PDF)

**S7 Fig. Accurate classification of roaming and dwelling at reduced sampling frequency.** A. Linear vs. angular velocity probability plot, calculated as described in Stern *et al.* (2017) [13] with centroid data from Stern *et al.* (2017) [13] Black dashed line shows split used to classify roaming vs. dwelling states. B. Linear vs. angular velocity probability plot, calculating angular velocity by using the Stern et al. (2017) [13] centroids at the current time as well as the centroid one minute in the past and one minute into the future. Black dashed line shows split at 90 degrees/min angular velocity used to classify roaming/ dwelling states with 79% accuracy based on ground truth classification in (A).
(PDF)

**S8 Fig. Accurate detection of worms using the WiCh model.** Histogram of IoU values for bounding boxes detected by the WiCh Faster R-CNN model compared to bounding boxes of hand annotated, segmented worms of the same frame. (n = 2550 frames).
(PDF)

**S9 Fig. Limitations with existing machine learning based segmentation tools.** A. Representative example frames of issues with segmentation using Ilastik within the same video. Even after training at least 50 frames (including a frame from the same video) the classification predictions and subsequent segmentations truncate the worm. Blue denotes background, yellow marks the worm, and red marks the egg objects. B. Representative example frames of issues with segmentation using Ilastik across similar videos. All frames were taken under the same imaging condition. (top) Prediction of pixel classification using the trained model. The model was trained with at least 50 images prior. Blue denotes background, yellow marks the worm, and red marks the egg objects. (bottom) Segmentation of objects based on the predictions. Note the truncation of worms and the misclassification of eggs as worms.
(PDF)

**S1 Table. Peak and average velocity for developmental data set and data from Stern *et al.* (2017) [13].**
(DOCX)

## Acknowledgments

The authors are grateful to Shay Stern, Cori Bargmann, Yuehui Zhao, and Patrick McGrath for generously providing video data, to Carys Thompson and Guillaume Aubry for testing the web-based pipeline, to Dhaval Patel for advice regarding *tph-1* animals and to QueeLim Ch'ng for strains.

## Author Contributions

**Conceptualization:** Kathleen Bates, Kim N. Le, Hang Lu.

**Data curation:** Kathleen Bates, Kim N. Le.

**Formal analysis:** Kathleen Bates, Kim N. Le.

**Funding acquisition:** Hang Lu.

**Investigation:** Kathleen Bates, Kim N. Le, Hang Lu.

**Methodology:** Kathleen Bates, Kim N. Le.

**Project administration:** Hang Lu.

**Resources:** Hang Lu.

**Software:** Kathleen Bates, Kim N. Le.

**Supervision:** Hang Lu.

**Validation:** Kathleen Bates, Kim N. Le.

**Visualization:** Kathleen Bates, Kim N. Le.

**Writing – original draft:** Kathleen Bates, Kim N. Le, Hang Lu.

**Writing – review & editing:** Kathleen Bates, Kim N. Le, Hang Lu.

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
