## [Decision Letter · Decision Letter 0]

13 Oct 2021

Dear Dr. Lu,

Thank you very much for submitting your manuscript "Deep learning for robust and flexible tracking in behavioral studies for C. elegans" for consideration at PLOS Computational Biology. As with all papers reviewed by the journal, your manuscript was reviewed by members of the editorial board and by several independent reviewers. The reviewers appreciated the attention to an important topic. Based on the reviews, we are likely to accept this manuscript for publication, providing that you modify the manuscript according to the review recommendations.

In particular please attend to the following points:

1) Although the reviewers were very positive about the fact that an attempt had been made to make the tools available for general use without a high technical bar, in practice, the reviewer who tried to use the provided software was unable to get it working. It is suggested that you actively test the ability of a third party to use the software and make any modifications required in the tools or instructions to ensure the this is straightforward.

2) Both reviews note that there is rather limited discussion of alternative approaches that use Deep Learning methods for animal tracking (such as Deep Lab Cut, LEAP). As mentioned in the reviews, sometimes these methods are being used for significantly different problems such as pose extraction vs. localization, but it would be helpful to make discussion of these differences more explicit. Nor is there reference to previous applications of the Faster R-CNN method to animal tracking, yet it has been used in several such applications to date, so providing more references and comparisons would be appropriate.

3) The reviewers note some situations in which the tracking seems less successful, and as they suggest, more discussion and analysis of the limitations of the method could be valuable.

Sincerely,

Barbara Webb

Associate Editor

PLOS Computational Biology

Dina Schneidman

Software Editor

PLOS Computational Biology

[LINK]

Reviewer's Responses to Questions

**Comments to the Authors:**

Reviewer #1: In this paper, Bates et al. describe a deep learning pipeline for identifying the locations of C. elegans L2-Adult and eggs in images. They show convincingly that it does an excellent job for this task despite variation in lighting conditions, decoys like worm tracks in the image background, and worm size. They illustrate its use on agar plates and in a microfluidic device, and for different purposes - fecundity assays, tracking movement, and growth measurements. They also have made the software available in an extremely easy to use form via well documented online jupyter notebooks off their github site.

First just a comment about software availability. Their web-based implementation of the software is just excellent, and it would be great if this became the standard expectation for this type of methods paper. All too often algorithms are described without any available implementation or there is an implementation but it is difficult to install or get to work or depends on extra software. By using online jupyter notebooks and open tools like colab, Bates et al. avoid these problems and make their software simple to use and access.

The paper is straightforwardly written and the results are well described. There are some things that could use some more explanation, however.

1) Lines 520-525 are really important! But they come at the very end. The method is framed as being able to "identify and detect worms" (eg. line 98). And of course it does that in a way. But it seems like it is more accurate to say that it identifies the _locations_ of worms. It identifies a bounding box that contains where a worm is but doesn't actually identify the worm's specific pixels. This is an important distinction for two reasons: (a) It means that this method is good for certain types of questions (line 518) but not others, and that it would need to be combined with something like Mask R-CNN (line 523) to give it that capability; (b) a lot of alternative methods that one might use (Supp fig 9) are actually focused on identifying the worm's pixels and so it's not really an apples-to-apples comparison. However, for any given application the comparison may be perfectly valid because if a bounding box is sufficient to answer the question then other methods are doing a worse job just to try to get extra information that's not needed. It's a good insight to realize that the location (and not the worm shape) is all you need to answer specific questions and that taking this distinction seriously makes it possible to develop methods like their WoP and WiCh that do this very well. But I didn't get this distinction until the very end after I'd spent the paper wondering why they were only looking at bounding boxes and not actually identifying worm pixels. I'd suggest adding a paragraph earlier making this point and also going into more detail about just why "the annotation necessary to train such a [segmentation] model is significantly more intensive." (line 525).

2) Because it identifies bounding boxes of worms, the orientation of the worms seems important. If a worm is oriented along the X or Y axis, then the bounding box will be thin and long. If it is oriented 45 degrees from that, then the bounding box will be square and have maximum area. This arbitrary orientation dependence seems like it would have an impact not just on where the centroid is but also on any overlap or area based metrics like loU in figure 4. As a suggestion, why not optimize the bounding box by rotating the image, identifying bounding boxes at each angle, and then picking the one with the smallest area (or some other metric like the intersection of all the boxes)? My speculation is that optimizing the orientation of the box might reduce some of the variability in some of the data (e.g. Fig S6A). Perhaps the intersection of all (or most) of the boxes might also be able to trim off pixels outside the convex hull of the worm. The downside would be that it would take longer to go through different angles, but there could be a way to do this in a way to cut down the number of angles tested - like a logarithmic search with the standard orientation and 45 degree rotation as starting points.

3) The paper doesn't deal with L1s or early L2s at all. Is this a limitation of their microscopy setup where the magnification was too low to get a lot of pixels for these worms? Or is it that because of background or whatever the software just doesn't work well for these worms and more magnification wouldn't help?

4) In figure 1B, the system doesn't seem to work well for a few of the worms towards the bottom which don't seem unusual conformation-wise (the ones in the middle, not the ones on the lower right which are clumped). What is going on there? What is it about these worms that makes them undetected? Along those lines, in the upper right there is a worm doing a turn that is not identified. Is this just a shortcoming of not enough turning worms in the training set or does the model have difficulty with worms of this shape?

5) The model seems like it does decently for identifying eggs in complex environments, but it is hard to tell where on the spectrum of typical plates their data fell. There are definitely worm tracks in the images in Figure 2A, but is this a typical background, and easy background, more worm tracks and intensity furrows that usual? It's hard to evaluate whether the numbers in lines 222 & 223 would be typical or not. However, the authors do appropriately make the comment that this egg detection is probably well suited for identifying trends (rather than precise counts) and so it's probably the case that identifying trends would be decently robust to image background variation.

Overall, this is a nice method, described in a well-written paper, provided in an excellent web-based format, and will be quite useful for a lot of worm assays. It makes an important implicit point (which should become explicit) that some kinds of information are easier to extract from images than others and that if this easier data is the essential data, you'd do better to focus on it and not try to do everything.

Minor points: Line 435: left -> top

Some of the methods have been described in previous papers (e.g. the microfluidic device, palmitic acid). Some are standard worm methods (culture on agar plates). The Raspberry Pi imaging setup is not cited, but neither is it described completely enough for someone to be able to replicate it. There is a mi-pi repository on the Lu lab github page. If this is the system used, I'd suggest adding a link around line 556.

Reviewer #2: While classic image processing techniques have enabled us to extract rich datasets that accurately describe the behavioural repertoire of C. elegans, there are still many behaviours that cannot be analysed using this approach. This leads to time consuming manual analysis and precludes larger scale studies. Bates et al., have utilised a deep learning approach to overcome this challenge. Using Faster R-CNN they trained and validated 3 models that have been tuned to look at 3 behaviours that are notoriously challenging to analyse in an automated way; development, egg laying, and aging. They have also established a web platform that allows other researchers to apply these models to their own data and train their own models. Their approach is also flexible and performs well under different environmental conditions, another important consideration for researchers who can't afford to set up a new imaging system. This work will be incredibly beneficial to many researchers who work with C. elegans.

I recommend the publication of this manuscript in PLOS Computational Biology; however, the authors need to first address the following comments.

1. I followed the ‘data annotation’ notebook until the ‘annotating images’ section and when I ran the cell I could see ‘loading widget…’ but no GUI emerged after several minutes. I didn’t test the training notebook as I got stuck at this point in the annotation notebook. I used the ‘Faster_R_CNN_inferencing’ notebook to test the egg model on data from our lab. I ran all the code and didn’t encounter any errors; however, no worms or eggs were detected. I then tested a video from the OpenWorm database to check if my data was the problem and again no worms were detected. I tried inputting the data as videos and as a folder of images and no worms were detected. I lowered both ‘target_min_scores’ values to 0.1 and still no worms were detected. I wondered if this was just an issue with the egg model, so I also tested the development model, and again no worms were detected. I may be doing something wrong or there is an invisible bug that’s preventing the model from being utilised correctly. I hope the authors can resolve this issue or provide additional instructions.

2. I think the authors should give a broader overview of deep learning approaches used for behavioural analysis in other species in the introduction. For example, DeepLabCut (http://www.mackenziemathislab.org/deeplabcut). Why did the authors choose Faster R-CNN over DeepLabCut?

3. Supplemental movie 1 shows the performance of the egg model on a video from the OpenWorm database. It appears the eggs are only counted after the worm has moved away from them and not immediately as they are laid (egg laying event). In many cases C. elegans researchers want to measure the time an egg is laid so that the interval between egg laying events can be analysed. I think the authors need to be more explicit about the limitations of their egg detection model for measuring an egg laying event.

4. Were all models trained on images from N2 worms? I’m wondering if the models will perform as well on worms with locomotion defects. Do the models take movement into consideration?

5. I really appreciate that the authors took the time to provide clear instructions and resources so that other scientists could train their own models. How do you suggest other scientists validate their models? I think the authors should discuss the limitations of deep learning approaches being used more broadly by non-experts without the correct validation.

6. The authors demonstrated how machine-learning image classification tools like Ilastik were not effective at segmenting the worms under different environmental conditions. Can the authors suggest any other deep learning segmentation approaches that might fair better? Or do they think deep learning is not the right approach for segmentation of worms?

7. Line 233 and 173: typo ‘lain’

**Have the authors made all data and (if applicable) computational code underlying the findings in their manuscript fully available?**

Reviewer #1: Yes

Reviewer #2: Yes

PLOS authors have the option to publish the peer review history of their article (what does this mean?). If published, this will include your full peer review and any attached files.

Reviewer #1: No

Reviewer #2: No

Figure Files:

Data Requirements:

Reproducibility:

References:

---

## [Decision Letter · Decision Letter 1]

21 Feb 2022

Dear Dr. Lu,

We are pleased to inform you that your manuscript 'Deep learning for robust and flexible tracking in behavioral studies for C. elegans' has been provisionally accepted for publication in PLOS Computational Biology.

Best regards,

Barbara Webb

Associate Editor

PLOS Computational Biology

Dina Schneidman

Software Editor

PLOS Computational Biology

Reviewer's Responses to Questions

**Comments to the Authors:**

Reviewer #1: The authors have addressed my previous comments. The discussion of the range of applicability and limitations of their approach versus other approaches are explained earlier and more clearly and have explained the false negatives in their images by (predominantly) a lack of similar images in their training set. The paper remains well-written, well-documented and should be of great interest to researchers who are thinking of applying CNNs to locate and track their organisms in images over time.

minor: line 94 there is a superfluous "with"

Reviewer #2: I think the authors have done a very good job addressing reviewer comments. I am supportive of publication.

**Have the authors made all data and (if applicable) computational code underlying the findings in their manuscript fully available?**

Reviewer #1: Yes

Reviewer #2: Yes

PLOS authors have the option to publish the peer review history of their article (what does this mean?). If published, this will include your full peer review and any attached files.

Reviewer #1: No

Reviewer #2: No

---

## [Editor Report · Acceptance letter]

3 Apr 2022

PCOMPBIOL-D-21-01379R1 

Deep learning for robust and flexible tracking in behavioral studies for *C. elegans*

Dear Dr Lu,

I am pleased to inform you that your manuscript has been formally accepted for publication in PLOS Computational Biology. Your manuscript is now with our production department and you will be notified of the publication date in due course.

With kind regards,

Olena Szabo
